# Forecasting the Bearing Capacity of the Driven Piles Using Advanced Machine-Learning Techniques

**Mohammed Amin Benbouras [1,2,\*]**, **Alexandru-Ionuţ Petrişor [3,4]**, **Hamma Zedira [5]**, **Laala Ghelani [5]** and **Lina Lefilef [6]**

1 Technology Department, École Normale Supérieure d'Enseignement Technologique de Skikda (ENSET), Skikda 21001, Algeria
2 Central Public Works Laboratory (LCTP), Algiers 16006, Algeria
3 Doctoral School of Urban Planning, "Ion Mincu" University of Architecture and Urbanism, 010014 Bucharest, Romania; alexandru.petrisor@uauim.ro
4 National Institute for Research and Development in Tourism, 50741 Bucharest, Romania
5 Civil Engineering Department, University of Abbes Laghrour, Khenchela 40051, Algeria; zedirahamma2003@yahoo.fr (H.Z.); ghilanilaala@yahoo.fr (L.G.)
6 Department of English Language and Literature, Mohamed Seddik Ben Yahia University, Jijel 18000, Algeria; lefileflina@gmail.com
\* Correspondence: mouhamed_amine.benbouras@g.enp.edu.dz

**Abstract:** Estimating the bearing capacity of piles is an essential point when seeking for safe and economic geotechnical structures. However, the traditional methods employed in this estimation are time-consuming and costly. The current study aims at elaborating a new alternative model for predicting the pile-bearing capacity based on eleven new advanced machine-learning methods in order to overcome these limitations. The modeling phase used a database of 100 samples collected from different countries. Additionally, eight relevant factors were selected in the input layer based on the literature recommendations. The optimal inputs were modeled using the machine-learning methods and their performance was assessed through six performance measures using a *K*-fold cross-validation approach. The comparative study proved the effectiveness of the DNN model, which displayed a higher performance in predicting the pile-bearing capacity. This elaborated model provided the optimal prediction, i.e., the closest to the experimental values, compared to the other models and formulae proposed by previous studies. Finally, a reliable and easy-to-use graphical interface was generated, namely "BeaCa2021". This will be very helpful for researchers and civil engineers when estimating the pile-bearing capacity, with the advantage of saving time and money.

**Keywords:** pile-bearing capacity; machine learning; deep neural network; *K*-fold cross-validation approach; sensitivity analysis

## 1. Introduction

Pile foundations are used to transmit construction loads deep into the ground in order to ensure structure stability [1,2]. Furthermore, computing the bearing capacity of piles is essential when designing economic and safe geotechnical structures [3]. To date, numerous approaches have been conceived for the sake of creating alternative methods and techniques that contain numerical, experimental, and analytical approaches aiming at predicting the bearing capacity of piles [4–6]. Among the most frequently used methods is the Cone Penetration Test (*CPT*), known for producing accurate results in a variety of situations [7,8]. This is probably due to the fact that *CPT*-based methods have been modeled in harmony with the *CPT* results, which were proven to estimate more effective different geotechnical properties, and make more precise pile capacity predictions [6]. Other semi-empirical methods have been widely utilized, such as Meyerhof's formula, which could yield an acceptable pile-bearing capacity [4]. On the other hand, the High-Strain Dynamic Load Test (*HSDLT*) and the Static Load Test (*SLT*) have been employed

considerably for predicting the pile-bearing capacity [9]. The *HSDT* is preferable to the *SLT*, because it operates with a faster, more advanced, and economic technology [2]. This quality supports its paramount importance addressed by the American Standards Test Methods to standardize the *HSDT* method [1]. The literature on bearing capacity values revealed a relatively close accuracy in both the *HSDT* and the *SLT* [1]. Momeni et al. [10] added that *HSDT* is faster and more economic compared to SLT, but it generally requires several *HSDT* tests for each project to obtain a reliable result [11]. Hence, increasing the number of *HSDT* tests is extremely undesirable since it may increase the total project budget. Moreover, other empirical researchers have proposed traditional methods for estimating the bearing capacity [12–15]. The quality of easiness and common usage has made these methods very important. However, determining the bearing capacity of bored and driven piles by means of the aforementioned methods is found to be time-consuming and costly [16]. This is probably due to the complex behavior of piles, heterogeneity of the soil around piles, material and shape of piles, and their installation. Accordingly, all the proposed methods/models in the literature yielded ineffective predictions [17]. On the other hand, currently, due to emerging new easy-to-use performance software such as PLAXIS, utilizing finite element analysis for which the system is discretized into a number of meshes to obtain axial capacity is of interest [18]. For this reason, numerical methods based on the finite element approach have recently become well-known for the evaluation of bearing capacity, yielding effective results [19,20]. Recently, the application of some new advanced techniques, namely "artificial intelligence (*AI*)" or "machine learning (*ML*)", has witnessed a spectrum of interest, and they provided exceptional results in solving several issues by learning from the available data [21,22].

Subsequently, the use of machine-learning methods to predict pile-bearing capacity has witnessed considerable development since the early 1990s [21–24]. Several studies are now able to estimate the pile-bearing capacity with a higher degree of precision in comparison to traditional methods. Among the fundamental research dealing with the pile-bearing capacity, Nawari et al. have used one hidden layer of the *ANN* model by investigating a database consisting of 25 test data. The chosen input parameters included the *SPT-N* values and geometrical properties. The *ANN* model efficiently predicted the pile-bearing capacity compared to traditional methods [25]. Furthermore, Mahnesh has predicted the pile-bearing capacity by using Support Vector Machines and Generalized Regression Neural Network with an input layer containing dynamic stress-wave data [26]. He concluded that the Generalized Regression Neural Network was the best model with a high correlation coefficient (0.977). In addition, Milad et al. have developed an effective model based on Artificial Neural Network, genetic programming, and linear regression methods to predict the bearing capacity of piles by learning from 100 samples. They utilized the Flap number, basic properties of the surrounding soil, pile geometry, and pile-soil friction angle as an input layer. The suggested *ANN* model has better stability compared to the other methods [27]. Jahed et al. used hybrid *PSO–ANN* to predict the bearing capacity of rock-socketed piles, by taking into consideration soil length to socket length ratio, total length to diameter ratio, uniaxial compressive strength, and standard penetration test. The proposed *PSO–ANN* model has demonstrated its efficiency since it produced a high correlation coefficient (R = 0.9685) [1]. Moayedi et al. have used *ANFIS*, *GP*, and *SA–GP* for modeling a database consisting of 50 tests. The chosen input parameters included the pile length, pile cross-sectional area, hammer weight, pile set, and drop height. The *SA–GP* model efficiently predicted the pile-bearing capacity compared to other methods [28]. Shaik et al. have predicted the pile-bearing capacity by using *ANFIS* and *ANFIS–GMDH–PSO* with an input layer containing *CPT* and pile loading test results [29]. They have proven that the metaheuristic hybrid *ANFIS–GMDH–PSO* model is the best one, with a high correlation coefficient (0.998) [29]. Harandizadeh et al. have used hybrid *MLP–GWO* and *ANFIS–GWO* to predict the bearing capacity of piles from the input layer, including pile area, pile length, flap number, average cohesion, and friction angle, average soil-specific weight, and average pile-soil friction angle. The proposed *MLP–GWO* model

has demonstrated that its efficiency yielded a high correlation coefficient ($R = 0.991$) [30]. Table 1 summarizes more than ten studies that have used machine-learning models to predict the pile-bearing capacity.

**Table 1.** Proposed machine-learning models in the literature to estimate the pile-bearing capacity.

| Authors | Inputs | Methods | Database | References |
|---|---|---|---|---|
| Nawari et al. (1999) | *SPT-N* values and geometrical properties | Neural Network | 25 | [25] |
| Mahnesh (2011) | Dynamic stress-wave data | Support Vector Machines and Generalized Regression Neural Network | 105 | [26] |
| Milad et al. (2015) | Flap number, basic properties of the surrounding soil, pile geometry, and pile-soil friction angle | Artificial Neural Network, Genetic Programming and Linear Regression | 100 | [27] |
| Jahed et al. (2017) | Soil length to socket length ratio, total length to diameter ratio, uniaxial compressive strength, and standard penetration test | hybrid *PSO–ANN* | 132 | [1] |
| Moayedi and Jahed (2018) | Internal friction angle of soil located in shaft and tip, pile length, effective vertical stress at pile toe and pile area | *ICA-ANN* | 59 | [31] |
| Yong et al. (2021) | Pile length, pile cross-sectional area, hammer weight, pile set, and drop height | *ANFIS, GP*, and *SA–GP* | 50 | [2] |
| Shaik et al. (2019) | Internal friction angle of soil located in shaft and tip, effective vertical stress at pile toe, pile area, and pile length | *ICA-ANN* and *ANFIS* | 59 | [29] |
| Kardani et al. (2020) | Shear resistance angle at the shaft of the pile, soil shear resistance angle at the tip of the pile, length of pile, cross-sectional area of the pile, and effective stress at the tip of the pile | *Decision tree, k-nearest neighbor, Multilayer Perceptron Artificial Neural Network, Random Forest, Support Vector Regressor*, and *Extreme Gradient Boosting* | 59 | [32] |
| Harandizadeh et al. (2021) | CPT and pile loading test results | *ANFIS* and *ANFIS–GMDH–PSO* | 72 | [30] |
| Moayedi et al. (2020) | Pile diameter, pile length, relative density, embedment ratio, and both the pile end resistance and base resistance | *GA-ANFIS* and *PSO-ANFIS* | 20 | [28] |
| Liu et al. (2020) | Laboratory and in situ testing results | *ANFIS, ANN*, and *GA-ANN* | 43 | [33] |
| Dehghanbanadaki et al. (2021) | Pile area, pile length, flap number, average cohesion and friction angle, average soil-specific weight, and average pile-soil friction angle | *MLP–GWO* and *ANFIS–GWO* | 100 | [34] |

According to the authors' knowledge, previous studies have been limited mostly to the use of *ANN, ANFIS,* and *SVM* methods for predicting the pile-bearing capacity, although recent studies have shown that other techniques could have yielded more effective and accurate results [35–37]. Furthermore, they assessed the predictive capability of suggested models depending on only one split to check the data learning validity. Consequently, the ability of their proposed model to overcome over-fitting and under-fitting problems cannot

be assured. Moreover, the majority of published papers have proposed machine-learning models in the form of mathematical equations, which are hard to duplicate in future studies. Admittedly, this practice has very little value for other researchers and civil engineers in the field. Conveniently, to overcome these limitations, investigators have presented their optimal models in the form of a programmed interface or a simple script by a well-known programming language such as Python or Matlab for generating the proposed model. This will make it available to anyone interested in the problem of modeling regardless of their proficiency level.

The current study contributes to providing a new alternative model for predicting the pile-bearing capacity based on 12 advanced machine-learning methods, which are applied for the first time for this aim. Furthermore, a high-performance method to estimate the generalization capability of the learning model, and to check the validity of the model for other cases, has been used, namely "*K*-fold cross-validation analysis". Finally, in order to treat the hard usage problem of machine-learning models in future studies, the proposed optimal model was used afterwards to develop a *GUI* public interface. Consequently, the suggested "BeaCa2021" interface is very handy and easy-to-use by civil engineers and researchers, by offering plenty of benefits such as reliability, easiness, and lowering the budget used to predict the pile-bearing capacity from relevant and easily obtained parameters without the need to operate expensive in situ tests.

## 2. Materials and Methods

### 2.1. Overview of the Methodology

Several advanced machine learning methods, such as Extreme Deep Neural Network (*DNN*), Extreme Learning Machine (*ELM*), Support Vector Regression (*SVR*), *LASSO* regression (*LASSO*), Random Forest (*RF*), Ridge Regression (*Ridge*), Partial Least Square Regression (*PLSR*), Stepwise Regression (*Stepwise*), Kernel Ridge (*KRidge*), Genetic Programming (*GP*), and Least Square Regression (*LSR*), have been used to learn from 100 samples collected from previous studies [27]. Multiple input parameters, including the pile material, average cohesion ($kN/m^2$), average friction angle (°), average soil-specific weight ($kN/m^3$), average pile-soil friction angle (°), flap number, pile area ($m^2$), and pile length (m), have been used. Firstly, the aforementioned advanced machine-learning methods have been utilized for modeling the input parameters, and their effectiveness was assessed through various statistical indicators. To evaluate the predictive ability of the optimal model, the *k*-fold cross-validation approach, which is based on five splits, has been employed. Afterward, in order to know which input variables have the biggest effect on the pile-bearing capacity through the proposed model, a sensitivity analysis has been performed via the step-by-step method. Finally, a reliable, easy-to-use, and the graphical interface was designed based on our optimal model in order to help civil engineers and researchers to easily predict the pile-bearing capacity in future studies.

### 2.2. Database

Choosing the Neural Network inputs is deemed to be the most significant phase for achieving accurate predictions. The selected relevant inputs should cover various aspects of the understudied problem. Besides, several factors have been selected, such as soil characteristics, pile-soil contact characteristics, and geometry and pile characteristics, which can affect the pile-bearing capacity. This study used data from 100 static load-bearing tests on the ultimate bearing capacity (*UBC*) of both the steel- and the concrete-driven piles from various countries, such as Iran, Mexico, and India [38–42]. The input parameters, including pile material, average cohesion ($kN/m^2$), average friction angle (°), average soil unit weight ($kN/m^3$), average pile-soil friction angle (°), flap number, pile area ($m^2$), and pile length (m), were selected as optimal input parameters. We have supposed that the cohesion, angle of shearing resistance, and soil unit weight were the parameters characterizing the soil condition, whereas the pile area and pile length are the parameters characterizing the pile geometric size. In addition, the pile-soil friction angle is the parameter describing the pile

material. Finally, the flap number was assumed to symbolize all other hidden effective factors in measuring the pile-bearing capacity [27]. The considered output was obtained from the static bearing capacity, which used static testing in the fully drained condition (long term). Two types of materials (concrete and steel piles, see Table S1) were used in this study. The data samples in both the training and validation phase have been randomly selected and completely detached. Table 2 shows the input and output parameters used in our study.

**Table 2.** Input and output parameters of the proposed model.

| Code | Parameter Type | Type of Variable | Subdivision | Variable |
|------|----------------|------------------|-------------|----------|
| X1 | Input | Qualitative | *X1* = 1 (Steel) <br> *X1* = 2 (Concrete) | Pile material |
| X2 | Input | Quantitative | | Average cohesion $(kN/m^2)$ |
| X3 | Input | Quantitative | | Average friction angle (°) |
| X4 | Input | Quantitative | | Average soil-specific weight $(kN/m^3)$ |
| X5 | Input | Quantitative | | Average pile-soil friction angle (°) |
| X6 | Input | Quantitative | | Flap number |
| X7 | Input | Quantitative | | Pile area $(m^2)$ |
| X8 | Input | Quantitative | | Pile length (m) |
| Y | Output | Quantitative | | Pile capacity (kN) |

*2.3. Machine-Learning Methods*

In the present paper, numerous machine-learning approaches have been utilized in order to perform a consistent study and to suggest an effective model. Many studies have revealed the effectiveness of the machine-learning methods, which have shown impressive results in the abroad fields. Hence, only the utilized methods are mentioned below, followed by some relevant references, which could be observed by the concerned readers to perfectly understand each method. The methods used were Deep Neural Network (*DNN*) [43,44], Extreme Learning Machine (*ELM*) [45], Random Forest (*RF*) [46], Support Vector Regression (*SVR*) [47], Partial Least Square Regression (*PLSR*) [48], LASSO regression (*LASSO*) [49], Kernel Ridge Regression (*KRidge*) [50], Ridge Regression (*Ridge*) [51], Genetic Programming (*GP*) [43], and Stepwise Regression (*Stepwise*) [52]. Matlab has been applied for modeling the algorithms corresponding to each method, except for GP, where the HeuristicLab Interface has been utilized [53]. The controlling parameters of the *ELM*, *DNN*, *SVR*, *RF*, *LASSO*, *PLS*, *Ridge*, *KRidge*, *Stepwise*, and *GP* algorithms used in this study are listed in Table 3. It is worth mentioning that the trial-and-error method has been applied in most *ML* approaches used in our study. This method is based on changing the controlling parameters of each technique and computing the mean square error in order to find the best parameters. Nevertheless, the controlling parameters of other methods, such as *ELM*, *PLS*, *Ridge*, and *KRidge*, are based on the aforementioned literature recommendations.

**Table 3.** Initial parameter settings for the algorithms.

| Algorithms | Algorithm Parameters | Value |
|---|---|---|
| ELM | Hidden layers<br>Hidden neurons<br>Activation function<br>Regulation parameter | $H = 1$<br>$N = 12$<br>'linear'<br>$C = 0.02$ |
| DNN | Hidden layers<br>Hidden neurons in the first layer<br>Hidden neurons in the second layer<br>Activation function in the first layer<br>Activation function in the second layer | $H = 2$<br>$N1 = [1\text{–}20]$<br>$N2 = [1\text{–}20]$<br>'Tansig'<br>'Tansig' |
| SVR | Regulation parameter $C$<br>Regulation parameter lambda<br>Kernel function | Series of $C$<br>Series of lambda<br>'rbf' |
| RF | nTrees<br>mTrees | nTrees = 100<br>mTrees = 26 |
| LASSO | Lambda | series of lambda |
| PLS | PLS components | NumComp = 3 for PSO<br>NumComp = 4 for GT and FS |
| Ridge | Regularization parameter lambda | lambda = 1 |
| KRidge | Regularization parameter lambda<br>Kernel function<br>Parameter for kernel | lambda = 1<br>'linear'<br>sigma = $2 \times 10^{-7}$ |
| GP | Function set<br>Population size<br>Number of generations<br>Genetic operators | $+, -, \times, \div$, power, ln, sqrt, sin, cos, tan<br>100 up to 500<br>1000<br>Reproduction, crossover, mutation |

*2.4. Statistical Performance Indicators*

The estimation precision of the suggested models was assessed through several statistical performance indicators and by utilizing graphical presentation. The statistical performance indicators are mean absolute error (*MAE*), root mean square error (*RMSE*), index of scattering (*IOS*), coefficient of determination ($R^2$), Pearson correlation coefficient (*R*), and index of agreement (*IOA*). They are expressed as follows [54,55]:

1.   Mean absolute error (*MAE*):

$$MEA = \frac{1}{N} \sum_{i=1}^{N} |Y_{tar,i} - Y_{out,i}| \quad (0 < MAE < \infty) \tag{1}$$

2.   Root mean square error (*RMSE*):

$$RMSE = \sqrt{\frac{1}{N} \sum_{i=1}^{N} (Y_{tar,i} - Y_{out,i})^2} \quad (0 < RMSE < \infty) \tag{2}$$

3.   Index of scattering (*IOS*):

$$IOS = \frac{\sqrt{\frac{1}{N} \sum_{i=1}^{N} (Y_{tar,i} - Y_{out,i})^2}}{\overline{Y_{tar}}} \quad (0 < RMSE < \infty) \tag{3}$$

4.   Coefficient of determination ($R^2$):

$$R^2 = 1 - \frac{\sum_{i=1}^{N}(Y_{tar,i} - Y_{out,i})^2}{\sum_{i=1}^{N}(Y_{tar,i} - \overline{Y_{tar}})^2} \quad (-\infty < NSE < 1) \tag{4}$$

5.　Pearson correlation coefficient (*R*):

$$R = \frac{\sum_{i=1}^{N}((Y_{tar,i} - \overline{Y_{tar}})(Y_{out,i} - \overline{Y_{out}}))}{\sqrt{\sum_{i=1}^{N}((Y_{tar,i} - \overline{Y_{tar}})^2(Y_{out,i} - \overline{Y_{out}})^2)}} \quad (-1 < R < 1) \tag{5}$$

6.　Index of agreement (*IOA*):

$$IOA = 1 - \frac{\sum_{i=1}^{N}(Y_{tar,i} - Y_{out,i})^2}{\sum_{i=1}^{N}\left(\sum_{i=1}^{N}|Y_{out,i} - \overline{Y_{tar}}| + \sum_{i=1}^{N}|Y_{tar,i} - \overline{Y_{tar}}|\right)^2} \quad (0 < IOA < 1) \tag{6}$$

where $Y_{tar,i}$, $Y_{out,i}$, $\overline{Y_{tar}}$, and $\overline{Y_{out}}$ characterize the target, output, mean of the target, and mean of output pile-bearing capacity values for *N* data samples, respectively. Moreover, the suggested machine-learning model possessed the minimum value of *RMSE*, *IOS*, and *MAE*, and the peak value of *IOA*, $R^2$, and *R* presents the optimal one and the closest to the experimental values.

Therefore, after choosing the optimal model based on statistical performance indicators, its predictive capability was evaluated by utilizing the *K*-fold cross-validation approach. The latter is an advanced approach, which revealed more accuracy and robustness when assessing the ability of the optimal model to overcome over-fitting and under-fitting problems in data learning [56,57]. The approach relies on dividing the database into *k* equal splits. Hence, for each split, *K*−1-folds are utilized for the training phase and the last one for validation. This procedure is reiterated successively until the use of all splits for the validation step [58,59]. The key benefit of this approach is that all the data are modeled in both the training and the validation steps [57]. Breiman and Spector have confirmed that *K* = 10- or *K* = 5-fold cross-validation is the best choice for assessing the model [56]. In our study, we selected *K*-fold cross-validation with *K* = 5 for assessing the predictive ability of the best model.

### 2.5. Methodology

In order to select the optimal model to predict the pile-bearing capacity using the aforementioned parameters as an input, the methodology followed the following phases:

1.　Creating a geotechnical database, collected from different countries such as Iran, Mexico, and India. In this step, 100 static load-bearing tests on the UBC of steel- and concrete-driven piles were collected as datasets.
2.　Modeling the chosen inputs by means of numerous machine-learning methods. The *ELM*, *DNN*, *SVR*, *RF*, *LASSO*, *PLS*, *Ridge*, *K Ridge*, *Stepwise*, and *GP* methods have been employed in this step for suggesting 11 models.
3.　Defining the optimal model for estimating the pile-bearing capacity value using important statistical performance indicators such as *MAE*, *RMSE*, *IOS*, $R^2$, *R*, and *IOA*.
4.　Evaluating the predictive capability of the optimal model to overcome under-fitting and over-fitting problems by utilizing the *K*-fold cross-validation approach with *K* = 5.
5.　Performing a sensitivity analysis by using the step-by-step method to define the most or least influential input on the bearing capacity via the proposed model.
6.　Designing a reliable, easy-to-use, and graphical interface based on our optimal model.

The research methodology for defining the optimal model to predict the pile-bearing capacity is systematically illustrated in Figure 1.

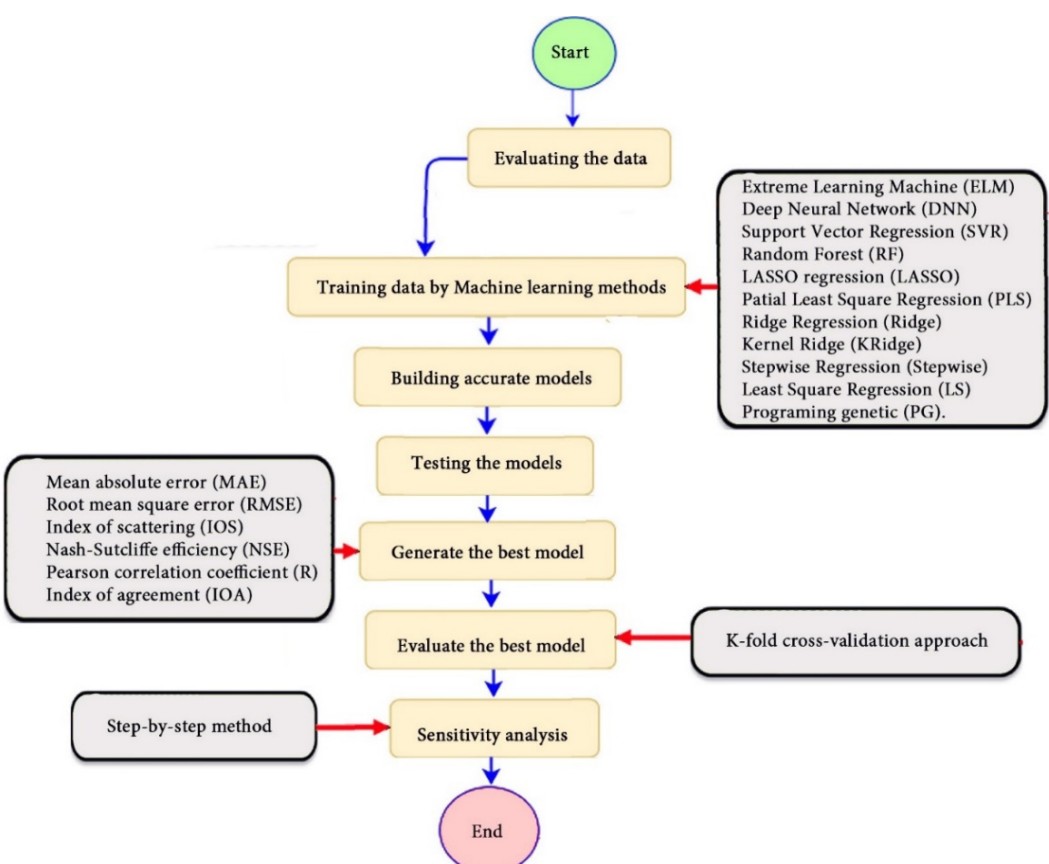

**Figure 1.** Flowchart describing the key steps for the methodology of research to estimate the pile-bearing capacity.

## 3. Results

### 3.1. Database Compilation

In the present paper, a database of 100 samples has been collected from previous studies, resulting in a dataset containing diverse data, considered as satisfactory for an efficient study. For the purpose of a precise modeling step, we have tried to make the dataset balanced for both concrete and steel material samples in both the training and validation phase. Furthermore, the data samples in both phases have been randomly chosen and completely detached. Table 4 shows the descriptive statistics of the user database, computed by using *SPSS*, including the range, minimum, maximum, mean, standard deviation (SD), variance, skewness, and kurtosis. The skewness values prove that all the parameters were equally distributed. Furthermore, the findings indicated that the dataset comprises a wide range of data. Consequently, the gathered database could be very handy when seeking to develop new empirical equations and models, as well as in evaluating the predictive capability of published formulae.

### 3.2. Correlation between Bearing Capacity and Input Parameters

To statistically estimate the relationship between the pile-bearing capacity and input parameters, SPSS software has been utilized. The correlation matrix between them is displayed in Figure 2, which shows a descriptive summary of the data distribution. The findings show a positive correlation between the pile-bearing capacity and other inputs, except for *X2*, *X4*, and *X5*, which appear to have a negative correlation (see Figure 2). This highlights that the decrease in these parameters tends to proportionally decrease the pile-bearing capacity. Moreover, Pearson correlation coefficient (*R*) and its significance between the pile-bearing capacity and other inputs is presented in Table 5. The findings prove that the significance is less than 0.05, except for *X3*, *X4*, and *X5*, showing that most correlations are statistically significant. Hence, according to Smith's classification (1986) [43], the pile-

bearing capacity is significantly correlated with the input parameters, excluding *X3*, *X4*, and *X5*, which are poorly correlated. The results point out that these factors can have a complex nonlinear relationship with the pile-bearing capacity. Besides, in order to precisely model this complex phenomenon, new sophisticated machine-learning approaches should be developed.

**Table 4.** Descriptive statistics of the collected samples (Std. Error = standard error, SD = standard deviation).

| | Range | Minimum | Maximum | Mean | | SD | Variance | Skewness | | Kurtosis | |
|---|---|---|---|---|---|---|---|---|---|---|---|
| | **Statistic** | **Statistic** | **Statistic** | **Statistic** | **Std. Error** | **Statistic** | **Statistic** | **Statistic** | **Std. Error** | **Statistic** | **Std. Error** |
| *X2* | 148.00 | 0.00 | 148.00 | 32.3741 | 3.28447 | 32.84 | 1078.77 | 2.011 | 0.241 | 4.570 | 0.478 |
| *X3* | 36.62 | 0.00 | 36.62 | 25.5803 | 0.96535 | 9.653 | 93.191 | −1.310 | 0.241 | 0.855 | 0.478 |
| *X4* | 8.11 | 5.38 | 13.49 | 10.2029 | 0.18409 | 1.840 | 3.389 | −0.406 | 0.241 | 0.262 | 0.478 |
| *X5* | 6.86 | 10.14 | 17.00 | 13.6823 | 0.16987 | 1.698 | 2.885 | 0.073 | 0.241 | −0.076 | 0.478 |
| *X6* | 2277.00 | 14.00 | 2291.00 | 494.99 | 60.23 | 602.32 | 362,794.16 | 1.502 | 0.241 | 1.286 | 0.478 |
| *X7* | 1.52 | 0.07 | 1.59 | 0.4327 | 0.04656 | 0.46562 | 0.217 | 1.128 | 0.241 | −0.233 | 0.478 |
| *X8* | 83.80 | 14.20 | 98.00 | 27.1120 | 1.86024 | 18.60 | 346.048 | 2.761 | 0.241 | 6.962 | 0.478 |
| *Y* | 51,560.00 | 540.00 | 52,100.00 | 5133.12 | 929.01 | 9290.14 | 86,306,843.19 | 4.043 | 0.241 | 16.258 | 0.478 |

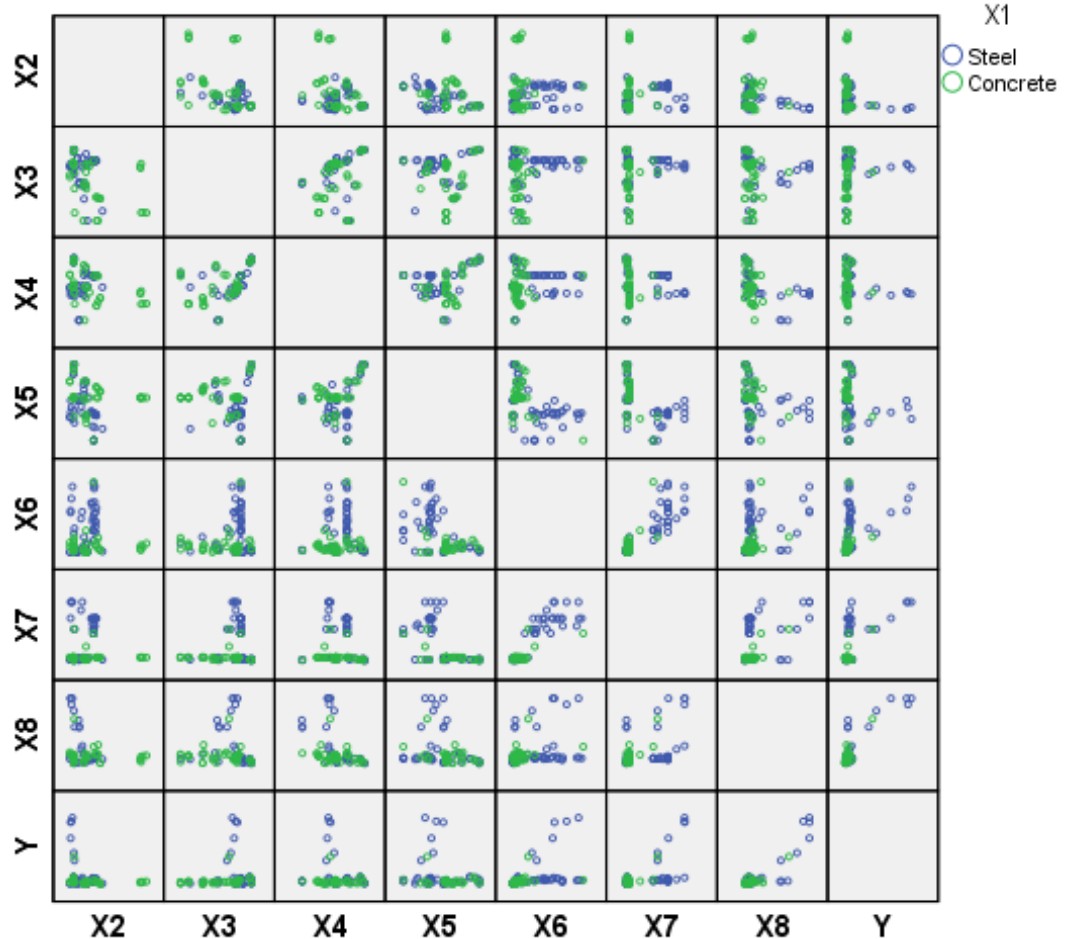

**Figure 2.** The correlation matrix between the pile-bearing capacity and soil parameters (green points: concrete material; blue points: steel material).

**Table 5.** Matrix of the correlation between the geotechnical parameters (**: Correlation is significant at the 0.01 level; *: Correlation is significant at the 0.05 level).

|  |  | *X2* | *X3* | *X4* | *X5* | *X6* | *X7* | *X8* | *Y* |
|---|---|---|---|---|---|---|---|---|---|
| *X2* | Pearson Correlation | 1 | −0.370 ** | −0.234 * | −0.221 * | 0.086 | 0.038 | −0.229 * | −0.229 * |
|  | Significance |  | 0.000 | 0.019 | 0.027 | 0.396 | 0.707 | 0.022 | 0.022 |
|  | N | 100 | 100 | 100 | 100 | 100 | 100 | 100 | 100 |
| *X3* | Pearson Correlation | −0.370 ** | 1 | 0.463 ** | 0.011 | 0.206 * | 0.259 ** | −0.063 | 0.099 |
|  | Significance | 0.000 |  | 0.000 | 0.916 | 0.040 | 0.009 | 0.531 | 0.326 |
|  | N | 100 | 100 | 100 | 100 | 100 | 100 | 100 | 100 |
| *X4* | Pearson Correlation | −0.234 * | 0.463 ** | 1 | 0.270 ** | 0.124 | 0.051 | −0.433 ** | −0.138 |
|  | Significance | 0.019 | 0.000 |  | 0.007 | 0.218 | 0.612 | 0.000 | 0.172 |
|  | N | 100 | 100 | 100 | 100 | 100 | 100 | 100 | 100 |
| *X5* | Pearson Correlation | −0.221 * | 0.011 | 0.270 ** | 1 | −0.489 ** | −0.555 ** | −0.189 | −0.142 |
|  | Significance | 0.027 | 0.916 | 0.007 |  | 0.000 | 0.000 | 0.059 | 0.159 |
|  | N | 100 | 100 | 100 | 100 | 100 | 100 | 100 | 100 |
| *X6* | Pearson Correlation | 0.086 | 0.206 * | 0.124 | −0.489 ** | 1 | 0.876 ** | 0.335 ** | 0.449 ** |
|  | Significance | 0.396 | 0.040 | 0.218 | 0.000 |  | 0.000 | 0.001 | 0.000 |
|  | N | 100 | 100 | 100 | 100 | 100 | 100 | 100 | 100 |
| *X7* | Pearson Correlation | 0.038 | 0.259 ** | 0.051 | −0.555 ** | 0.876 ** | 1 | 0.446 ** | 0.563 ** |
|  | Significance | 0.707 | 0.009 | 0.612 | 0.000 | 0.000 |  | 0.000 | 0.000 |
|  | N | 100 | 100 | 100 | 100 | 100 | 100 | 100 | 100 |
| *X8* | Pearson Correlation | −0.229 * | −0.063 | −0.433 ** | −0.189 | 0.335 ** | 0.446 ** | 1 | 0.866 ** |
|  | Significance | 0.022 | 0.531 | 0.000 | 0.059 | 0.001 | 0.000 |  | 0.000 |
|  | N | 100 | 100 | 100 | 100 | 100 | 100 | 100 | 100 |
| *Y* | Pearson Correlation | −0.229 * | 0.099 | −0.138 | −0.142 | 0.449 ** | 0.563 ** | 0.866 ** | 1 |
|  | Significance | 0.022 | 0.326 | 0.172 | 0.159 | 0.000 | 0.000 | 0.000 |  |
|  | N | 100 | 100 | 100 | 100 | 100 | 100 | 100 | 100 |

On the other hand, we were generally interested in the correlation between inputs if the multicollinearity phenomenon existed. This could appear between certain independent variables with a high $R$, causing problems when fitting the model and interpreting the results, and reducing the statistical power of the regression model. However, the correlation coefficient presented in Table 5 indicates a moderate $R$ between inputs, indicating a moderate multicollinearity, but it is not severe enough to require corrective measures. For this reason, there was no interest in the correlation coefficient between input variables.

### 3.3. Bearing Capacity Prediction through AI Models

To define the optimal machine-learning model, the first step consists of selecting the optimal input parameters that have a high influence on the target value, and the second step is to determine the best machine-learning methods. To begin with, in order to define the suitable input parameters, eight factors have been used following the literature recommendations. Afterward, we attempted to determine the optimal *ANN* model for predicting the pile-bearing capacity depending on six statistical measures. The performance of each model for the selected optimal input in both concrete and steel piles is presented in Table 6. Six performance measures have been used to compare the proposed models in order to select the best one, in terms of the mean absolute error (*MAE*), root mean square error (*RMSE*), index of scattering (*IOS*), coefficient of determination ($R^2$), Pearson correlation coefficient ($R$), and index of agreement (*IOA*). The data were divided into two parts, i.e., 80% for the training and 20% for the validation. As Table 6 demonstrates, the target values were modeled via the machine-learning methods, where the parameters of the methods have been fixed (as presented in Table 4) and compared using the six performance measures in order to find the best model. The different models produced the values: *MAE* ($0.1650 \times 10^3$ to $3.0424 \times 10^3$), *RMSE* ($0.2140 \times 10^3$ to $4.2390 \times 10^3$), *IOS* (0.0755 to 0.7737), $R$ (0.9315 to 0.9977), $R^2$ (0.8676 to 0.9954), and *IOA* (0.9360 to 0.9988) in concrete piles. Similarly, in the steel piles, we obtained *MAE* ($0.1870 \times 10^3$ to $3.1064 \times 10^3$), *RMSE* ($0.3100 \times 10^3$ to $4.3966 \times 10^3$), *IOS* (0.0448 to 0.9081), $R$ (0.8478 to 0.9997), $R^2$ (0.7187 to 0.9994), and *IOA* (0.9118 to 0.9998). The results indicate that the best performance was obtained from the *DNN* model trained by the Tan-Sigmoid function. This model is said to be the most appropriate one because it displays the highest accuracy in terms of *MAE* ($0.1650 \times 10^3 / 0.1870 \times 10^3$), *RMSE* ($0.214 \times 10^3 / 0.31 \times 10^3$), *IOS* (0.0755/0.0448), $R$ (0.9977/0.9997), $R^2$ (0.9954/0.9994), and *IOA* (0.9988/0.9998) in both concrete/steel piles. Finally, the most appropriate *DNN* model displayed the higher values of performance measures criteria in both the training and validation phase. Furthermore, this model is closely followed by the *GP* model, which shows an acceptable accuracy as it ranked second. Moreover, the results showed the poor performance of the *ELM* model in predicting the pile-bearing capacity. With respect to the performance of machine-learning models during the training phase, the performance hierarchy follows the following order: *DNN, GP, RF, Kridge, SVR, LS, Ridge, Step, PLS, Lasso,* and *ELM*. Finally, the scatter plots between the target and the output bearing capacity value of each model are presented in Appendix A (Figures A1–A11).

### 3.4. Evaluating the Best Fitted Model Using the K-Fold Cross-Validation Approach

The 5-fold cross-validation approach was effectively utilized to evaluate the predictive capability of the optimal model. It is worthy to note that the aforementioned studies interested in predicting the pile-bearing capacity have assessed the predictive capability of their optimal models based on one single split. Consequently, the ability of the models to overcome the over-fitting and under-fitting problems could not be verified. Figure 3 displays the performance measures of the optimal *DNN* models utilizing 5-fold cross-validation based on the validation data for each split. The results clearly indicate the fulfillment of the *DNN* model. Additionally, the fact that the correlation coefficient ranged between 0.9777 and 0.9998 for data validation in the 5 splits proved the predictive capability

of the optimal *DNN* model to learn existing data, generate novel validation data, and overcome over-fitting and under-fitting problems.

**Table 6.** Performance indicators values of the AI models for predicting the pile-bearing capacity in both concrete and steel piles (bold: the optimal model).

|  | $MAE \times 10^3$ | $RMSE \times 10^3$ | $IOS$ | $R$ | $R^2$ | $IOA$ |
|---|---|---|---|---|---|---|
| Concrete piles |  |  |  |  |  |  |
| DNN | 0.1650 | 0.2140 | 0.0755 | 0.9977 | 0.9954 | 0.9988 |
| ELM | 3.0424 | 4.2390 | 0.7737 | 0.9320 | 0.8686 | 0.9610 |
| Lasso | 2.4324 | 3.5390 | 0.6637 | 0.9620 | 0.9254 | 0.9700 |
| PLS | 2.5524 | 3.6390 | 0.6837 | 0.9688 | 0.9386 | 0.9700 |
| RF | 1.1024 | 2.1690 | 0.3837 | 0.9880 | 0.9761 | 0.9912 |
| Kridge | 2.2930 | 3.5917 | 0.6816 | 0.9433 | 0.8899 | 0.9641 |
| Ridge | 2.4268 | 3.6145 | 0.6876 | 0.9409 | 0.8853 | 0.9636 |
| LS | 2.3093 | 3.5867 | 0.6824 | 0.9414 | 0.8863 | 0.9656 |
| Step | 2.4738 | 3.6421 | 0.6970 | 0.9352 | 0.8746 | 0.9626 |
| SVR | 1.9787 | 4.0984 | 0.7734 | 0.9315 | 0.8676 | 0.9360 |
| GP | 0.5966 | 0.9612 | 0.1731 | 0.9975 | 0.9951 | 0.9961 |
| Steel piles |  |  |  |  |  |  |
| DNN | 0.1870 | 0.3100 | 0.0448 | 0.9997 | 0.9994 | 0.9998 |
| ELM | 3.1064 | 4.3966 | 0.9081 | 0.8478 | 0.7187 | 0.9118 |
| Lasso | 2.7149 | 3.6962 | 0.7527 | 0.8990 | 0.8082 | 0.9437 |
| PLS | 2.6329 | 3.6973 | 0.7763 | 0.8966 | 0.8038 | 0.9398 |
| RF | 1.1213 | 2.3475 | 0.4893 | 0.9875 | 0.9751 | 0.9712 |
| Kridge | 2.2482 | 3.6937 | 0.7342 | 0.8993 | 0.8088 | 0.9441 |
| Ridge | 2.3820 | 3.7165 | 0.7402 | 0.8969 | 0.8044 | 0.9436 |
| LS | 2.2646 | 3.6887 | 0.7350 | 0.8974 | 0.8054 | 0.9456 |
| Step | 2.4291 | 3.7441 | 0.7496 | 0.8912 | 0.7943 | 0.9426 |
| SVR | 1.9340 | 4.2004 | 0.8260 | 0.8875 | 0.7876 | 0.9160 |
| GP | 0.5518 | 1.0632 | 0.2257 | 0.9975 | 0.9951 | 0.9965 |

### *3.5. Comparison between the Proposed Models and Empirical Formulae*

To test the effectiveness of the suggested *DNN* model, a comparative study was performed using 12 empirical models proposed in the literature of predicting the bearing capacity, as presented in Table 7. It should be noted that no author has shared the mathematical equations of the proposed *ML* model to compare results with the same database. Published research was limited in presenting modeling results. Therefore, we cannot validate the proposed models using the current collected dataset. Consequently, the current study was limited to compare the proposed models based on the correlation coefficient. It is needless to say that the correlation coefficient is an important indicator when assessing the prediction precision, as the best model is represented by a prediction value close to 1. The results of the comparative study indicated that the proposed *DNN* model in our study is the best-performing model, with maximum accuracy (0.9996 for all data). Furthermore, our model is closely followed by the *ANN* model which was proposed by Milad et al. [27], and it showed an acceptable accuracy as it ranked secondly. Moreover, the results revealed the poor performance of the ANN model proposed by Nawari et al. [25] in the bearing capacity. With respect to the performance of machine-learning models, the hierarchy follows the

following order: Milad et al. [27], Liu et al. [32], Yong et al. [28], Moayedi et al. [34], De-hghanbanadaki et al. [30], Mahnesh [26], Kardani et al. [2], Jahed et al. [1], Shaik et al. [33], Moayedi and Jahed [31], Harandizadeh et al. [29], and Nawari et al. [25]. We believe that the reasonable ground standing behind the high accuracy found in our suggested model is due to deep learning (more than one hidden layer). The latter could offer the necessary flexibility for modeling complex functions in many cases.

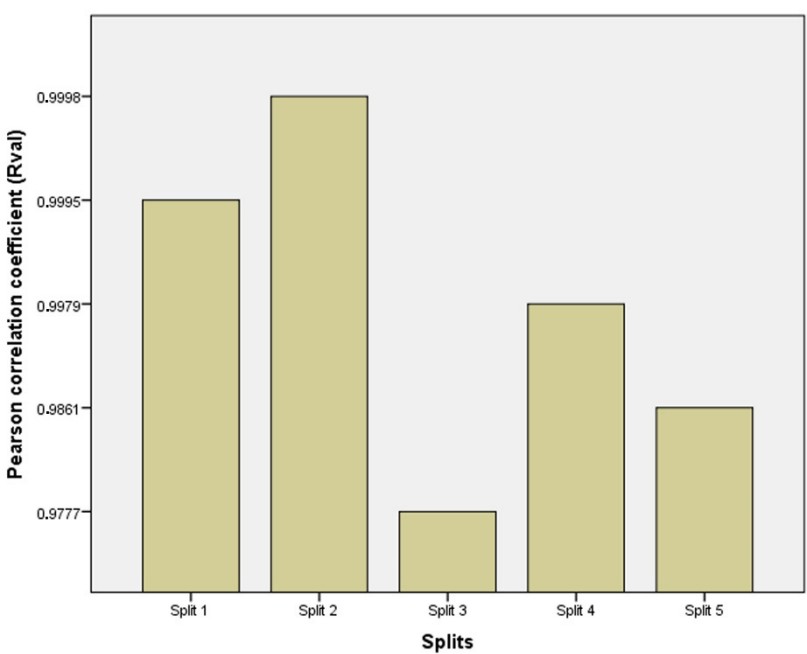

**Figure 3.** Performance measures of the DNN model using the *K*-fold cross-validation, with *K* = 5.

**Table 7.** Comparison between the proposed DNN model and some of the empirical models found in the literature.

| Authors | Sample Size | Best Methods | Correlation Coefficient | References |
|---|---|---|---|---|
| Nawari et al. (1999) | 25 | *ANN* | 0.91 | [25] |
| Mahnesh (2011) | 105 | Generalized Regression Neural Network | 0.977 | [26] |
| Milad et al. (2015) | 100 | Neural Network | 0.9995 | [27] |
| Jahed et al. (2017) | 132 | *PSO–ANN* | 0.9685 | [1] |
| Moayedi and Jahed (2018) | 59 | *ICA-ANN* | 0.96369 | [31] |
| Yong et al. (2021) | 50 | *GP* | 0.997 | [28] |
| Shaik et al. (2019) | 59 | *ANFIS* | 0.967 | [33] |
| Kardani et al. (2020) | 59 | Extreme Gradient Boosting | 0.975 | [2] |
| Harandizadeh et al. (2021) | 72 | *ANFIS–GMDH–PSO* | 0.94 | [29] |
| Moayedi et al. (2020) | 20 | *GA–ANFIS* | 0.9935 | [34] |
| Liu et al. (2020) | 43 | *GA-ANN* | 0.998 | [32] |
| Dehghanbanadaki et al. (2021) | 100 | *MLP–GWO* | 0.991 | [30] |
| Our study | 100 | Deep Neural Network | 0.9996 | |

### 3.6. Sensitivity Analysis

In order to know what input variables have a significant effect on the pile-bearing capacity, with the assistance of the DNN model, a sensitivity analysis was performed by utilizing the step-by-step technique [60]. In this method, each normalized input parameter varies at a constant rate, one at a time, while the other variables are held constant. Diverse constant rates (0.3, 0.6, and 0.9) were chosen in this study. For every input, the percentage of variation in the output, as a result of the variation in the input, was computed. The sensitivity of each input was computed based on Equation (7):

$$\text{Sensitivity level of } X_j (\%) = \frac{1}{K} \sum_{i=1}^{K} \left( \frac{\% \ change \ in \ output}{\% \ change \ in \ input} \right)_i \tag{7}$$

where *K* refers to the number of the datasets used in the study (*K* = 100). The outcomes of the sensitivity analysis of the proposed *DNN* model are illustrated in Figure 4. It can be noticed that the pile-bearing capacity was significantly influenced by the pile area, and its sensibility ratio ranged between 26.3% and 38.06%. This parameter is closely followed by the pile length, which showed a moderate sensitivity level that ranged between 15% and 19%. In addition, the cohesion and friction angle had a moderate effect on the pile-bearing capacity, with a sensibility ratio ranging between 9% and 15%. Finally, other parameters had little effect on the target values.

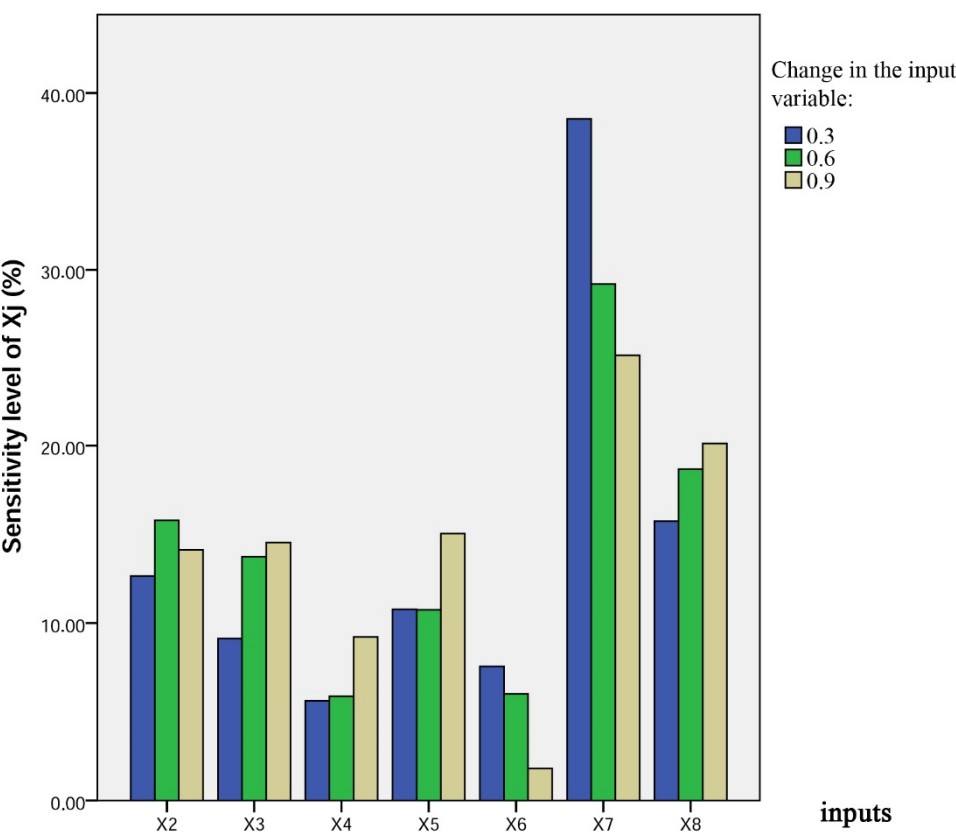

**Figure 4.** Results of the sensitivity analysis of the proposed model.

### 3.7. Graphical User Interface (GUI) Design "BeaCa2021"

It is a common practice in the majority of published papers using machine-learning modeling to present models in the form of mathematical equations, which suffer from their hard fitting in future studies. Seemingly, this practice has very little value for other researchers and civil engineers in the field. In order to make it useful, the proposed machine-learning architecture should be presented either in the form of a programmed interface

such as Matlab or in a simple script employing a known programming language such as Python for generating the proposed model [55]. In such a case, the machine-learning model can be readily used and is thus available to anyone interested in the problem of modeling. In this study, a reliable, graphical, and easy-to-use interface was designed based on our optimal *DNN* model, as presented in Figure 5. The proposed optimal model was afterward used to develop a *GUI* public interface. The designed interface, called "BeaCa2021", was programmed by Matlab software. The reason for choosing this name is due to "Bea" relative to "Bearing", "Ca" relative to Capacity, and 2021, the year this interface was designed. In addition, BeaCa2021 includes the most relevant input parameters on the bearing capacity. Initially, the user must define the pile material type (either steel or concrete). Secondly, the user is required to introduce the other input parameters: average cohesion, average friction angle, average soil-specific weight, average pile-soil friction angle, flap number, pile area, and pile length. Finally, by clicking Run, the prediction result appears in the outputs. The suggested BeaCa2021 interface will be very useful to civil engineers and researchers, by helping them to predict the bearing capacity, which is deemed as one of the most complex parameters to determine.

**Figure 5.** BeaCa2021 interface.

## 4. Discussion

In the current study, a very important contribution in the geotechnical community has been introduced for the sake of enhancing the performance of the pile-bearing capacity model. It is worth mentioning here that the model quality is influenced by the method utilized. Hence, other unused advanced machine-learning methods demonstrated efficient results in other areas. Consequently, in the current study, we examined the usage of twelve advanced machine-learning methods, such as Deep Neural Network (*DNN*), Extreme Learning Machine (*ELM*), Support Vector Regression (*SVR*), LASSO regression (*LASSO*), Random Forest (*RF*), Ridge Regression (*Ridge*), Partial Least Square Regression (*PLS*), Stepwise Regression (Stepwise), Kernel Ridge (*KRidge*), Genetic Programming (*GP*), and Least Square Regression (*LSR*), to predict the pile-bearing capacity. According to the authors' knowledge, the use of the aforementioned machine-learning methods in predicting

the pile-bearing capacity is very rare. Therefore, this study began with collecting a wide range of data consisting of 100 static load-bearing tests on the *UBC* of both steel- and concrete-driven piles from different countries, such as Iran, Mexico, and India. Afterward, we selected eight relevant factors based on the literature recommendations, such as average cohesion (kN/m$^2$), average friction angle (°), average soil-specific weight (kN/m$^3$), average pile-soil friction angle (°), flap number, pile area (m$^2$), and pile length (m). Based on that, eleven advanced machine-learning methods (*DNN*, *ELM*, *SVR*, *LASSO*, *RF*, *Ridge*, *PLS*, *Stepwise*, *KRidge*, *GP*, and *LS*) were applied for modeling the selected optimal input set for the first time. The findings clearly indicate that the Deep Neural Network (*DNN*) presents the most appropriate model, which yielded the minimum values of error metrics (*MEA*, *RMSE*, and *IOS*) and the higher values of $R^2$, *R*, and *IOA* compared to other models. Furthermore, the newly developed model was assessed by the *K*-fold cross-validation method and compared to other proposed models from the literature based on the correlation coefficient. The conclusion drawn is that the optimal *DNN* model could produce new data without causing over-fitting or under-fitting, plus being much more precise than the other proposed empirical models. Moreover, the last part in the current study consisted of the sensitivity analysis, which provided an overview of the most influential parameters on the pile-bearing capacity according to the proposed model. The findings indicate that the pile area was the most influential factor on the pile-bearing capacity. Pile length also had a considerable effect. In addition, the cohesion and friction angle demonstrated a moderate effect on the pile-bearing capacity, with a sensibility ratio ranging between 9% and 15%. Finally, the proposed optimal model was then used to develop a *GUI* public interface in order to facilitate its usage in the future. A reliable, easy-to-use, and graphical interface, named "BeaCa2021, presented in the current study, was programmed via Matlab software. The essential advantage of "BeaCa2021" is to help researchers and civil engineers interested in the problem of modeling regardless of their proficiency, by offering them plenty of benefits, such as reliability, easiness, and lowering the budget used for predicting the pile-bearing capacity from relevant and easily obtained parameters without the need to operate expensive in situ tests.

The results obtained in the current study also proved that the performance of the pile-bearing capacity model was considerably enhanced by using new machine-learning methods. The model prediction by the *DNN* was improved by 8.91% with the *ANN* method proposed by Nawari et al. [25], 3.58% with the *PSO–ANN* method proposed by Jahed et al. [1], and 0.86% with the *MLP–GWO* method proposed by Dehghanbanadaki et al. [30]. The obtained results are logical because deep learning is generally employed either in the prediction or in the problematic classification, which can reduce the bias and variance plus avoiding over-fitting and under-fitting problems, as opposed to the traditional *ANN* methods, to improve their predictive capability. According to these data, we can infer that the *DNN* method, which was employed in this study for the first time for the purpose of modeling the pile-bearing capacity, could yield more effective and accurate results than the other machine-learning methods.

Despite the multiple extraordinary findings of this study, a number of important limitations need to be addressed. The fundamental limitation would be the fact that the sample size was relatively small, which may affect the precision of the pile-bearing capacity. This may lead to the proposed model's inability to generalize the new conditions or circumstances that were not used in the training data stage. Besides, researchers generally utilize large and diverse data collected by transferring knowledge between them. This is an important issue to build on in future research, i.e., to rely on the data gathered from multiple countries to enhance its learning and, therefore, produce a better model. Additionally, further studies using meta-heuristic algorithms for the prediction of pile-bearing capacity are strongly recommended. We mention, for example, the Particle Swarm Optimization (*PSO*) and Gravitational Search Algorithm (*GSA*), Bee Colony Algorithm (*BCA*), Bio-geography-Based Optimization (*BBO*), Whale Optimization Algorithm (*WOA*), Ant Colony Optimization (*ACO*), and Grey Wolf Optimizer (*GWO*). These algorithms have

shown high-performance results when combined with machine-learning techniques, leading to improving their learning, and therefore rapidly converging to the best solution. The application of these meta-heuristic algorithms combined with machine-learning methods has shown impressive results in the abroad fields [55,61].

## 5. Conclusions

This study relied on a considerable number of steel- and concrete-driven pile data collected from different countries, such as Iran, Mexico, and India. The comparison of the results' assessment between the different proposed models revealed the superiority of the *DNN* model proposed in our study, which yielded the highest accuracy in terms of *MAE*, *RMSE*, *IOS*, *R*, $R^2$, and *IOA* in both the training/validation phases. The findings indicate that this model has a high correlation coefficient, ranging between 0.9777 and 0.9998 for the validation data in the 5 splits of the *k*-fold cross-validation approach, meaning that there was no over-fitting or under-fitting. Furthermore, the results indicated that the aforementioned *DNN* model is more effective compared to other empirical models proposed in the literature. The sensitivity analysis results proved that pile area had the most significant effect on the prediction of the pile-bearing capacity. Pile lengths had a moderate influence and were ranked second. In addition, cohesion and friction angle had little effect on the pile-bearing capacity. Finally, the proposed optimal model was then used to develop a *GUI* public interface with Matlab software, named "BeaCa2021". The fundamental benefit of "BeaCa2021" is to help researchers and practicing civil engineers, regardless of their proficiency, interested in the problem of modeling, to estimate the pile-bearing capacity with the benefits of gaining time and money.

This work has opened up several questions that need further investigations to overcome certain limitations. Firstly, there is a need to use more data from other countries to enhance the learning phase, which is needed to develop the BeaCa2021 in the future. Secondly, we propose the usage of meta-heuristic algorithms combined with machine-learning methods for predicting the pile-bearing capacity in future studies. These algorithms have demonstrated high-performance results when used with machine-learning techniques, leading to improved learning.

**Supplementary Materials:** The following are available online at https://www.mdpi.com/article/10.3390/app112210908/s1, Table S1: The Database about Two Types of Materials.

**Author Contributions:** Conceptualization, M.A.B.; methodology, M.A.B. and H.Z.; software, M.A.B. and L.G.; validation, M.A.B. and A.-I.P.; formal analysis, M.A.B. and L.L.; investigation, M.A.B.; resources, M.A.B.; data curation, M.A.B.; writing—original draft preparation, M.A.B., L.L., and A.-I.P.; writing—review and editing, M.A.B., L.L., and A.-I.P.; visualization, M.A.B. and A.-I.P.; supervision, M.A.B., H.Z., and A.-I.P.; project administration, M.A.B. and L.G.; funding acquisition, M.A.B. and A.-I.P. All authors have read and agreed to the published version of the manuscript.

**Funding:** This research received no external funding.

**Institutional Review Board Statement:** Not applicable.

**Informed Consent Statement:** Not applicable.

**Conflicts of Interest:** The authors declare no conflict of interest.

## Appendix A

The scatter plots between target and output pile-bearing capacity values by the advanced machine-learning models.

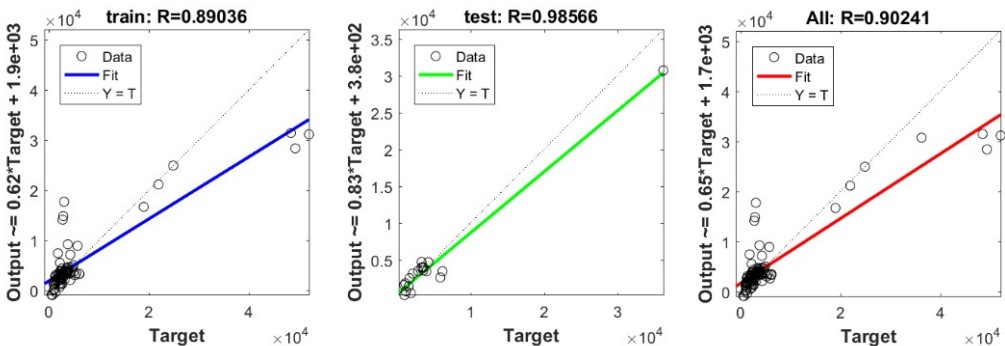

**Figure A1.** Scatter plots between target and output values by the *SVR* model.

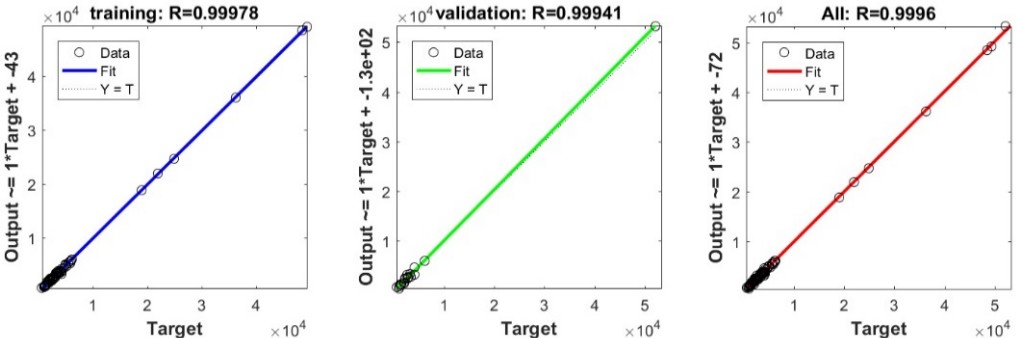

**Figure A2.** Scatter plots between target and output values by the *DNN* model.

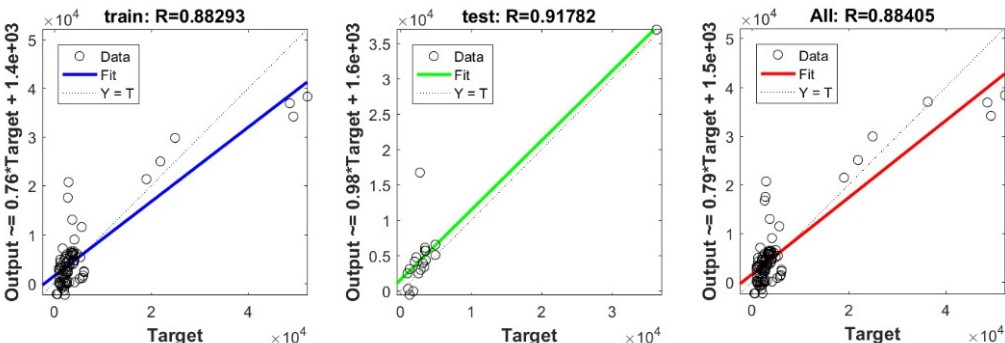

**Figure A3.** Scatter plots between target and output values by the *ELM* model.

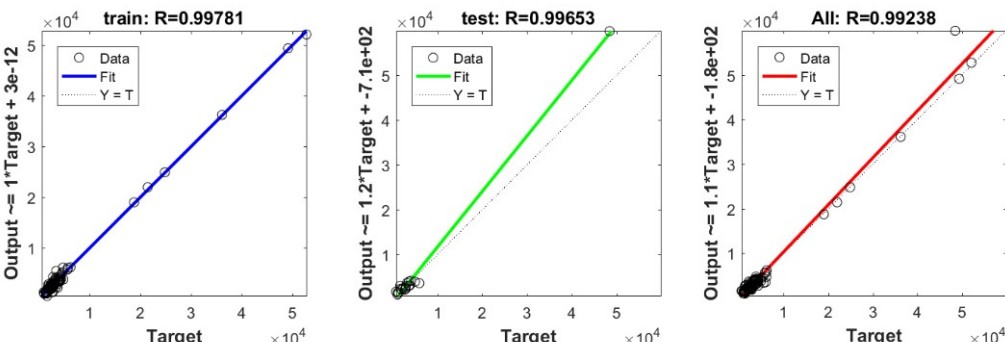

**Figure A4.** Scatter plots between target and output values by the *GP* model.

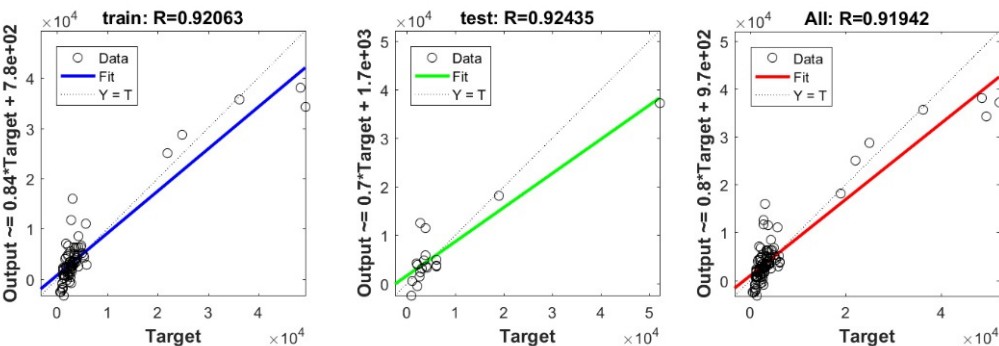

**Figure A5.** Scatter plots between target and output values by the *KRidge* model.

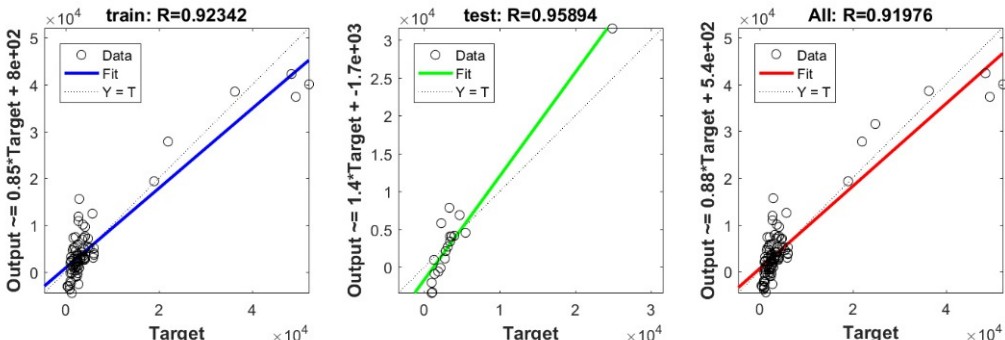

**Figure A6.** Scatter plots between target and output values by the *LASSO* model.

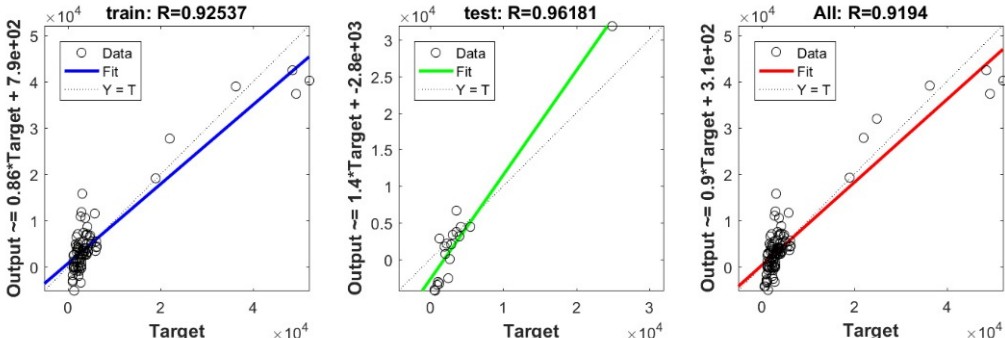

**Figure A7.** Scatter plots between target and output values by the *PLS* model.

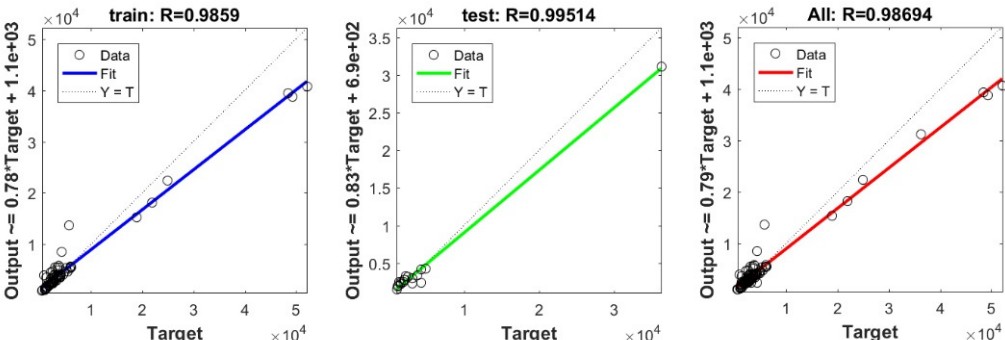

**Figure A8.** Scatter plots between target and output values by the *RF* model.

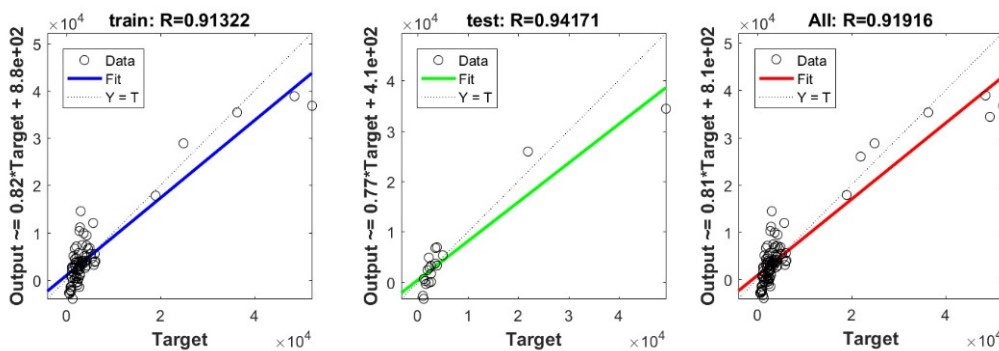

**Figure A9.** Scatter plots between target and output values by the *Ridge* model.

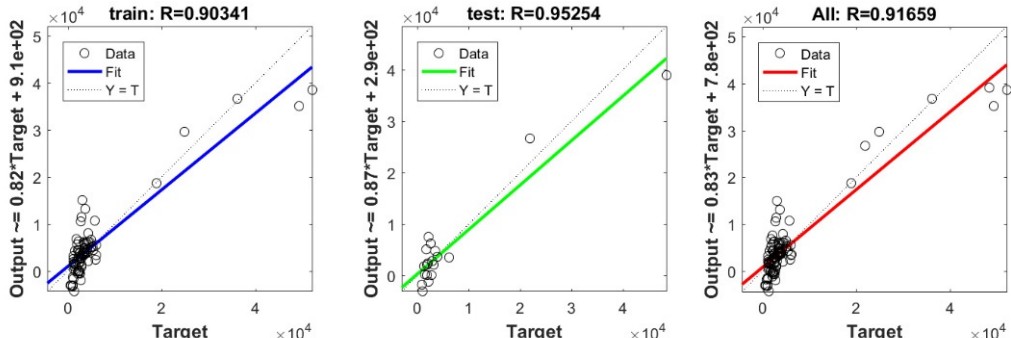

**Figure A10.** Scatter plots between target and output values by the *STEP* model.

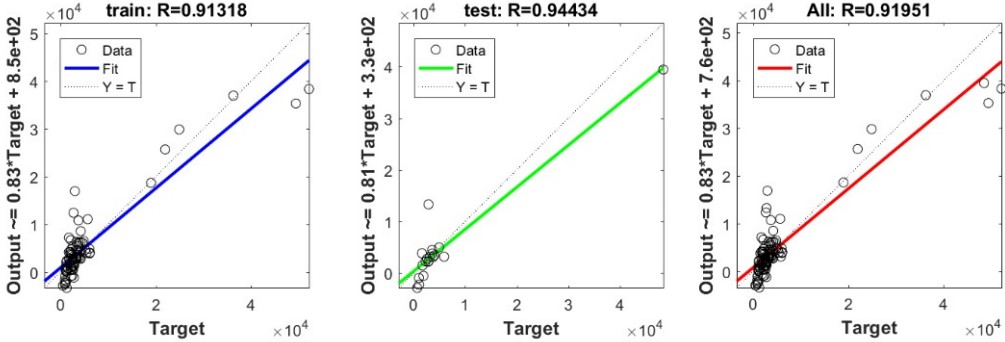

**Figure A11.** Scatter plots between target and output values by the *LS* model.

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
