# Peer review of "Forecasting the Bearing Capacity of the Driven Piles Using Advanced Machine-Learning Techniques"

_applsci, doi:10.3390/app112210908_

Round 1

Reviewer 1 Report

This paper compared several machine-learning (ML) techniques for pile capacity prediction based on a database of 100 samples and determined the best performed technique accordingly. A graphical interface within Matlab was then proposed in order to facilitate the application of the selected ML technique in practice. The idea is interesting especially the graphical interface. However, the study quality is not good enough and there is a lack of novelty comparing some existing papers. As a result, a decision of ‘reject’ is recommended. 

My concerns about the study quality and novelty:

  1. My biggest concern is that the compared ML techniques were not optimized. Some default parameters (not explained in the paper) were used, so the comparison could be unfair (some techniques use their optimal parameters for the comparison but the others cannot). Usually, evolutionary algorithms are used to optimize the hyper-parameters of ML techniques and different database may lead to different optimal parameters.
  2. According to the figure in Appendix A, the values of Y are mainly lower than 1E4; only a small part of the values are higher than 2E4. The measured performance of each technique is a quality representation from a global viewpoint (i.e., for the range [0, 5E4]). Whether the technique performs well within small values (i.e., 1E4) is thus not clear. In fact, it is better to separate the data of concrete and steel piles and conduct two separated comparisons (one for concrete piles and one for steel piles), since they have different characteristics as evidenced in Figure 2 (the correlation between input variables and green points is significantly different from the one of variables – blue points).
  3. The novelty of the present study is not enough as there exists some similar works in literature (some examples are given as follows). The contribution is not significant compared to these works.

[1] Kardani N, Zhou A, Nazem M, et al. Estimation of bearing capacity of piles in cohesionless soil using optimised machine learning approaches[J]. Geotechnical and Geological Engineering, 2020, 38(2): 2271-2291.

[2] Moayedi H, Moatamediyan A, Nguyen H, et al. Prediction of ultimate bearing capacity through various novel evolutionary and neural network models[J]. Engineering with Computers, 2020, 36(2): 671-687.

My questions and other comments:

  1. Is the method applicable for the piles installed within multi-layered soils?
  2. Better to give the definition of every abbreviation such as UBC.
  3. In the considered database, how many samples are for concrete piles?
  4. Table 4: what ‘Std. Error’ stands for?
  5. Page 8: ‘the pile bearing capacity is evenly correlated with the input parameters’. Please explain why the pile capacity is EVENLY correlated with the input parameters?
  6. There is no comment on the correlation between any two input parameters.
  7. Table 5: ‘Sig. (2-tailed)’?
  8. Section 3.5: it seems that the authors compared different methods by only regarding the correlation coefficient which was obtained with different databases. The comparison is thus unfair. It is better to use the methods recently proposed in literature and compare them with the DNN model of the paper by using SAME databases.

Some typo errors

  1. Page 5: … in abroad fields, Hence, …
  2. Page 7: … and India.In this step …
  3. Page 12: others parameters

Author Response

Thank you for your time, efforts and suggestions. Details are available in the attached file.

Reviewer 2 Report

The abstract represents the essence of the paper. The presented study aims to elaborate a new alternative model for pile bearing capacity prediction based on eleven new advanced machine learning methods. A database of 100 samples from different countries was used in the modelling phase. In addition, eight relevant factors were selected for the input layer based on literature recommendations. The optimal inputs were modelled using machine learning methods and their performance was evaluated using six performance measures with a K-fold cross-validation approach. The comparative study proved the effectiveness of DNN model which showed higher performance in predicting the pile bearing capacity. This elaborated model provided the optimum prediction, i.e., the one closest to the experimental values, compared to the other models and formulas proposed in previous studies.

Introduction is clear, but the last part of the introduction is more like a conclusion - “The present study contributes to providing a reliable, simple, and easy-to-use interface for predicting the pile bearing capacity, named “BeaCa2021”. Firstly, several advanced Machine Learning methods that have not been used previously, were utilized in practical modeling aiming at predicting the pile bearing capacity. Secondly, the capability of the optimal model to overcome the over-fitting and under-fitting problems has been tested through the K- cross-validation approach. Finally, the proposed optimal model was afterward used to develop a GUI public interface. Consequently, the suggested “BeaCa2021” interface will be very handy and easy-to-use by civil engineers and researchers, by offering plenty of benefits such as reliability, easiness, and lowering the budget used to predict the pile bearing capacity from relevant and easily- obtained parameters without the need to operate expensive in-situ tests.”

Materials and Methods

The methodology for selecting the optimal model for pile bearing capacity prediction using the above parameters as inputs is divided into different phases and clearly presented.
Several advanced machine learning methods such as Extreme Deep Neural Network (DNN), Extreme Learning Machine (ELM), Support Vector Regression (SVR), LASSO regression (LASSO), Random Forest (RF), Ridge Regression (Ridge), Partial Least Square Regression (PLSR), Stepwise Regression (Stepwise), Kernel Ridge (KRidge), Genetic Programming (GP) and Least Square Regression (LSR) were used to learn from 100 samples from previous studies. Multiple input parameters were used, including the Pile Material, Average Cohesion (kN/m2), Average Friction angle (°), Average soil Specific weight (kN/m3), Average Pile-Soil friction angle between pile and soil (°), Flap Number, Pile Area (m2), and Pile Length (m). Initially, the advanced machine learning methods mentioned above were used to model the input parameters; their effectiveness was evaluated using various statistical indicators. To evaluate the predictive ability of the optimal model, the k-fold cross-validation approach based on 5 splits was used. Then, to identify the input variables that have the most influence on the bearing capacity of the pile by the proposed model, a sensitivity analysis was performed using the step-by-step method. Finally, based on our optimal model, a reliable, easy-to-use graphical interface was developed to help civil engineers and researchers predict the bearing capacity of piles in future studies.

Results
A database of 100 samples from previous studies was compiled.
The results are described. The data in the tables and all figures are clearly presented.
The comparative study proved the effectiveness of DNN model which showed higher performance in predicting the bearing capacity of the piles. This elaborated model provided the optimum prediction, i.e., the one closest to the experimental values, compared to the other models and formulas proposed in previous studies. Finally, a reliable and easy-to-use graphical interface was created, namely "BeaCa2021". This will be very helpful for researchers and civil engineers in estimating the bearing capacity of piles and has the advantage of saving time and money.

Conclusion
This work has raised several issues that need further investigation. Firstly, there is a need to use more data from other countries to improve the learning phase required for the future development of BeaCa2021. Second, the use of meta-heuristic algorithms in combination with machine learning methods to predict the bearing capacity of piles is proposed in future studies. These algorithms have shown powerful results when used with machine learning methods, leading to an improvement in the learning process.

Author Response

(The authors gave the same response as above.)

Reviewer 3 Report

The manuscript deals with the evaluation of the bearing capacity of pile foundations through the artificial intelligence technique. By reading the manuscript, it appears that this approach would be alternative and better than other approaches commonly employed in practice and research (in the field of civil / geotechnical engineering). However, contrary to the above mentioned methods, the proposed approach is not based on any equation of physical meaning. Rather, it seems more based on statistics and the results should be strongly affected by the employed database. Actually, the fact that a database is required for the calculation of the bearing capacity of a pile is a limitation itself. As a consequence, a method such that proposed in the present study could be aimed at most for a preliminary evaluation of the bearing capacity, and is not alternative to more rigorous approaches. This aspect should be clearly highlighted in the manuscript.

Consequently, the manuscript presents some shortcomings in the present form and a major revision is required before it can be further considered for possible publication in the journal. All details are reported in the following.

Required changes:

  1. Despite understandable, English needs some improvements.
  2. Originality/novelty of the study proposed. This issue is very important and should be better clarified and well highlighted in the text.
  3. Lines 15-16: “However, the traditional methods employed in this estimation are less effective, time-consuming, and costly”. Probably, geotechnical engineers would not agree with this statement. Indeed, the traditional numerical methods for the calculation of the bearing capacity of pile foundations are very simple to use, i.e. they are not time consuming and not costly. Besides, they should be less effective than what?
  4. Lines 26-27: the evaluation of the bearing capacity of pile foundations (as other problems of geotechnical engineering) is a problem of paramount importance in the field of civil engineering and follows specific physical rules. Therefore, traditional methods cannot be overtaken by artificial intelligence. In this context, the ‘graphical interface’ generated in this study could be employed in practice at most for a preliminary evaluation of the bearing capacity. However, the design of a civil engineering structure cannot rely on such a calculation and must be verified with a more rigorous approach.
  5. Introduction: the literature review provides some example of methods for the calculation of the bearing capacity of pile foundations. However, most of the cited studies (with the exception of Meyerhof [4]) are quite ‘unknown’. Some of the most traditional and commonly employed methods among practitioners are missing and should be added to the references (for example: De Beer, 1945; Brinch Hansen, 1951; Berezantsev, 1965; Terzagni, 1943; Vesic 1977). Furthermore, methods based on the finite element approach have also become common recently (Achmus and Thieken, 2010; Conte et al., 2021; Graine et al., 2021).
  6. Line 153: since soil is a multiphase material, the term specific weight is not correct. Rather, it is generally indicated as ‘unit weight’. Analogously, friction angle should be ‘angle of shearing resistance’.
  7. Lines 224-226: a dataset of 100 case studies was collected. However, all the data concerning the 100 cases should be reported (values of input and output parameters). Furthermore, how was the bearing capacity of these cases evaluated?
  8. Section 3.7: it is not clear how this BeaCa2021 works. What equations does it solve?

REFERENCES

Achmus, M., and K. Thieken. 2010. “On the behavior of piles in non-cohesive soil under combined horizontal and vertical loading.” Acta Geotech. 5 (3): 199–210. https://doi.org/10.1007/s11440-010-0124-1.

Berezantsev, V.G. (1965) Design of deep foundations, Proceedings of the Sixth International Conference of Soil Mechanics and Foundation Engineering, Vol. 2, Montreal, pp. 234–237.

Brinch Hansen, J. (1951) Simple Statical Computation of Permissible Pile Load, CN-Post, Copenhagen.

Conte, E.; Pugliese, L.; Troncone, A.; Vena, M. 2021. "A simple approach for evaluating the bearing capacity of piles subjected to inclined loads." International Journal of Geomechanics (ASCE), 21(11): 04021224. DOI: 10.1061/(ASCE)GM.1943-5622.0002215.

De Beer, E. (1945) Etude des fondations sur piloitis et des fondations directes, Annales des Travaux Publiques de Belgique, 46: 1–78.

Graine, N., M. Hjiaj, and K. Krabbenhoft. 2021. “3D failure envelope of a rigid pile embedded in a cohesive soil using finite element limit analysis.” Int. J. Numer. Anal. Methods Geomech. 45 (2): 265–290. https://doi.org/10.1002/nag.3152.

Terzaghi, K.(1943) Theoretical Soil Mechanics, John Wiley, New York.

Vesic, A.S. (1977) Design of Pile Foundations: National Cooperative Highway Research Program, Synthesis Highway Practice Report No. 42, Transport Research Board, Washington, DC.

Author Response

(The authors gave the same response as above.)

Round 2

Reviewer 1 Report

I appreciate the efforts made by the authors to answer my questions.

My comments are as follows:

Comment 1: It is important to clarify whether EACH of the comparted ML techniques is optimized with respect to a same database (e.g., the database in hand) or not. It seems that the parameters of some techniques are taken from published papers based on different databases (usually with a small number of samples). Ideally, we need to optimize the parameters of each method to be compared at first with a same database, and then compare the optimal version of each method in order to find the best one for the database. If the size of the database is large, we could have more confidence to generalize the determined best model.

In the current study, the database size is not large (around 50 samples for respectively concrete and steel piles) and not all the methods are optimized with the database in hand. This is why I have concerns about the fairness of the comparison and the generalization of the conclusions.

Comment 2: Only the results of DNN are presented in Table 11. Have the authors compared DNN with other ML techniques for separated databases of concrete and steel piles? As the two types of piles have different characteristics, the best ML technique could be different for them (not sure, but better to show the study).

The study of Table 10 is not useful from my viewpoint. The results in this table show only the global (averaged) performance of each method considering both the concrete and steel piles. Table 10 should be replaced by two tables respectively for the two types of piles.

Comment 8: According to the classifications of Smith (1986), the pile bearing capacity is moderately correlated with all the input parameters. I am still confused with the word ‘evenly’. why evenly?

Comment 9: please add/adapt the Response to the manuscript as readers need to know why you present the correlation coefficients between the input variables in Table 5 and Figure 2.  

Comment 11: please clearly mention the limitations of the comparison of section 3.5 in the manuscript.

Author Response

(The authors gave the same response as above.)

Reviewer 3 Report

The replies provided by the authors are sufficient to address the previous comments. Accordingly, the manuscript could be accepted in the present form.

Author Response

Thank you for your time, efforts and suggestions.

Round 3

Reviewer 1 Report

Thanks for the revision.

I have only one comment.

Comment 1: please mention in the paper that the trial/error method was used to optimize most of the used ML techniques while the parameters of other techniques (which one?) are taken from published papers (if I understand well).

This is important for researchers who would like to repeat or improve your study.

Author Response

(The authors gave the same response as above.)
